# Retrospective Analysis of the Psychological Predictors of Public Health Support in Bulgarians at the Beginning of the Coronavirus Pandemic

**DOI:** 10.3390/brainsci13050821

**Published:** 2023-05-19

**Authors:** Kristina Stoyanova, Drozdstoy Stoyanov, Angel M. Dzhambov

**Affiliations:** 1Research Institute at Medical University of Plovdiv, Research Group “Translational and Computational Neuroscience”, Strategic Research and Innovation Program for Development of MU-Plovdiv, Medical University of Plovdiv, 4002 Plovdiv, Bulgaria; drozdstoy.stoyanov@mu-plovdiv.bg; 2Department of Psychiatry and Medical Psychology, Medical University of Plovdiv, 4002 Plovdiv, Bulgaria; 3Department of Hygiene, Faculty of Public Health, Medical University of Plovdiv, 4002 Plovdiv, Bulgaria; angel.dzhambov@mu-plovdiv.bg; 4Research Group “Health and Quality of Life in a Green and Sustainable Environment”, Strategic Research and Innovation Program for Development of MU-Plovdiv, Medical University of Plovdiv, 4002 Plovdiv, Bulgaria; 5Institute of Highway Engineering and Transport Planning, Graz University of Technology, 8010 Graz, Austria

**Keywords:** psychological predictors, public health support, psychological well-being, conspiracy theories beliefs, precarity, ontological uncertainty, affective polarization

## Abstract

The earliest critical context of the pandemic, preceding the first real epidemiological wave of contagion in Bulgaria, was examined using a socio-affective perspective. A retrospective and agnostic analytical approach was adopted. Our goal was to identify traits and trends that explain public health support (PHS) of Bulgarians during the first two months of the declared state of emergency. We investigated a set of variables with a unified method within an international scientific network named the International Collaboration on Social & Moral Psychology of COVID-19 (ICSMP) in April and May 2020. A total of 733 Bulgarians participated in the study (67.3% females), with an average age of 31.8 years (SD = 11.66). Conspiracy Theories Beliefs were a significant predictor of lower PHS. Psychological Well-Being was significantly associated with Physical Contact and Anti-Corona Policy Support. Physical Contact was significantly predicted by fewer Conspiracy Theories Beliefs, higher Collective Narcissism, Open-mindedness, higher Trait Self-Control, Moral Identity, Risk Perception and Psychological Well-Being. Physical Hygiene compliance was predicted by fewer Conspiracy Theories Beliefs, Collective Narcissism, Morality-as-Cooperation, Moral Identity and Psychological Well-Being. The results revealed two polar trends of support and non-support of public health policies. The contribution of this study is in providing evidence for the affective polarization and phenomenology of (non)precarity during the outbreak of the pandemic.

## 1. Introduction

Belief in conspiracies in the age of COVID-19 were distinctly prevalent. They were updated on multiple themes such as wearing masks, vaccines, the idea of the coronavirus as a biological weapon, oligarchic and government interventions with financial and economic motives, etc. [1]. Research has recently proposed a social-functional model of conspiracy beliefs in which precarity is a central psychosocial construct. Adam-Troian et al. [2] argue that precarity expands the social-psychological lore of conspiratorial attitudes as an indirect consequence of structural issues, such as social inequalities. They define precarity as the subjective experience of persistent social and psychological insecurity within objective conditions of affiliation and economic deprivation. Perspective for precarity as a motivation of the conspiracy mentality, related as ontological uncertainty and existential threat [3,4,5], is inherently affective. The way one projects oneself into the future suffers. A theoretical pole of that theory is ontological security, which implies fundamental senses of safety and trust sustaining to individual psychological well-being. According to the authors, people perceive conspiracy narratives through an already shaped basic sense of (dis)trust [2]. In this way, the idea is articulated that contexts actualize affective human nature based on personal phenomenological experience.

The critical epidemic realities narrowed psychosocial functioning to a common pattern of public health regulation in all countries that have been affected by SARS-CoV-2. In turn, this pattern has opened up space for polarization in different societies [6,7,8,9]. While political polarization is better outlined, socio-psychological polarization is more delicate to explore. Some authors have analyzed cognitive rigidity and neglecting alternative information in the interpretation of fake and real news as socio-cognitive polarization [10]. Intragroup antagonistic tendencies in political and ideological contexts have been described as affective polarization. It has been established that affective polarization is related to phenomena of agonistic democracy and anti-democratic attitudes [11,12]. There is evidence that empathic concern, a latent trait of empathy, increases levels of affective polarization [13]. Research on the relation between personality traits and affective polarization is actually insufficient and has concentrated on bias behavior and political preferences [14,15,16,17]. Personality traits explain the cognitive reading of reality, but group dynamics are moderated by universal variables such as identity and belonging [18,19,20]. We assume that the social-affective framework is conceptually the most syncretic for understanding the background of the current investigation, as described in the sections to follow.

The measures to protect against the viral invasion of COVID-19 were simple and universal—physical distancing, physical hygiene and long-term adherence to policies limiting all forms of group contact [21]. However, human functioning in its integrative sense is not simple and universal. Personality is an individual organization of a psychobiological system (body, thoughts, psyche), within which a person modulates their experience and adapts to an ever-changing internal and external environment [22,23]. Research on the psychobiological model of personality has operationalized well-being as an implicit variable of the functioning of human beings. More specifically, a tripartite structure of subjective well-being is investigated—positive affect, negative affect and life satisfaction, and these components are analyzed and assessed both independently and jointly [24]. Thus, the contribution of individual traits (temperament and character traits) to people’s adaptive functioning reflected in well-being is well-established. Subjective well-being encompasses cognitive and emotional aspects of subjective feelings regarding individual life circumstances. Individual differences in positive affect, negative affect and life satisfaction are explained by different organizations of psychobiological systems and processes [24]. These differences correspond to three distinct systems of human learning and memory described as associative conditioning, intentional self-control and self-awareness. Recent results show that negative affect and life satisfaction are dependent on a personality network for intentional self-control, and positive affect is dependent on a personality network for self-awareness [25].

Global research established that conspiracy beliefs were associated with low adherence to anti-epidemic public health guidelines. Conspiracy theory beliefs also mediate a negative relation between national narcissism and engagement in public behaviors [26]. Within the ICSMP population survey was found a positive association between Conspiracy Theories Beliefs and dimensions of Moral Identity and Morality-as-Cooperation. This finding was interpreted as a dilemma in people’s moral judgment that determines their behavior with regard to public health [27]. Moral identity is associated with commitment, meaning, identification with and acceptance of others (Cooperativeness) and with feeling that one is part of something bigger than oneself (Self-transcendence) [28]. Internalized moral identity was the most consistent predictor of attitudinal and behavioral responses to COVID-19. Morality-as-Cooperation was associated with behavioral responses, most consistently in predicting hygiene maintenance. Open-mindedness and self-control were positively associated with avoiding contact and supporting policy, and Open-mindedness was interpreted as an aspect of cognitive humility or the readiness to accept information contrary to initial beliefs. Social Belonging predominantly predicted hygiene maintenance. Collective narcissism was a predictor of political support and contact avoidance [29].

The low public health support (PHS) assessment raised the issue of distrust in public institutions such as health systems and governments in the pandemic time. Negative perception of the surrounding social environment expressed in an assessment of unreliable social structures and people was related to a lack of institutional trust and the sense of an unreliable context constraint identification with government measures and decisions [30]. The belief that the health threat of coronavirus is exaggerated predicted both non-compliance with public health recommendations and reduced support for government actions against the virus’s spread [31]. Concerns about unreliable or insufficient online information regarding coronavirus spread and prevention were undermining support for health protection rules, while fear related to work and personal health motivated people’s support [32]. Recent research found that core belief violation and meaning-making, as well as more intense perceptions of vulnerability and mortality, mediate the impact of pandemic stressors on mental health. Worse mental health state was explained not by the objectively traumatic nature of pandemic-related events, but by their perception as existentially threatening. Specifically, job loss or reduction was a stressor that predicted both higher anxiety and better mental health indicators. Processes of attribution of meaning have given rise to these polar differences in emotional state [33].

We have theoretical and objective reasons to consider these results in the affective perspective of precarity and ontological certainty of adaptation and subjective well-being. With a particular reference to the Bulgarian context, recent studies of well-being and values-based mental health studies indicate the controversies and compromises which underpin mental health in the view of the cultural pluralism in Bulgaria [34,35]. With the psychobiological paradigm as an explanatory model, we argue that, in collective behavior, the individual organization of the psychobiological system is always revealed in an affective configuration of adaptive and maladaptive modalities. Our hypothesis is that conspiracy theories, wellbeing and personality traits contribute to affective polarization and precarity in the public context. The following constructs from the overall panel of ICSMP measures were selected to be relevant to that hypothesis: Conspiracy Theories Beliefs, Collective Narcissism, Open-mindedness, Trait Self-Control, Moral Identity, Risk Perception and Psychological Well-Being.

## 2. Materials and Methods

### 2.1. Participants and Procedure

An online survey in Bulgarian was conducted that was part of the International Collaboration on Social & Moral Psychology of COVID-19 (ICSMP) [36,37]. Within two months from the beginning of April to the end of May 2020, the survey was active and distributed through an administrative online link to which our research team had regular permanent access. A Google form was created which was addressed on behalf of the Medical University of Plovdiv to its employees, students and their families, to the wider community and to institutional partners. The introductory section of the online form was unified according to project policy. It contained a summary of the purpose of the study, informed consent options and approval from the Research Ethics Committee, University of Kent, United Kingdom, No. 202015872211976468. Additionally, we received institutional support for conducting the research from the Medical University of Plovdiv within the framework of a currently active national project, named “COVID-19 HUB—Information, Innovations and Implementation of Integrative Scientific Developments” in the thematic area of medical-biological problems, financed by the Bulgarian National Science Fund under contract No. KP-06-DK1/6 dated 29 March 2021. All constructs were validated within the ICMP project itself.

Sampling was conducted using the snowball method. The Bulgarian sample included 794 individuals. After data cleaning, 733 Bulgarian participants were included in this study. We excluded surveys that were incomplete for any reason. The likely explanation is that people opted out of finalizing their participation at some point. The main inclusion criteria were that the survey was finalized and that a respondent was 18 years of age to declare informed consent and voluntary participation. We translated into Bulgarian the original English text of the survey using the forward-backward method. Essentially, the instrument can be seen as a battery of self-assessment tests. Then, we dedicated time to an independent psycholinguistic evaluation of the Bulgarian text to ensure feasibility of items toward Bulgarian cultural attitudes. To that end, we conducted an online pilot survey with feedback from respondents, which informed further revisions to the text. This turned out to be critically important, as the timing of research historically preceded the several epidemiological waves of the pandemic and the impact of world statistics in terms of mortality and infection with coronavirus later. We conceptually tested both Bulgarian-specific and culture-independent psychological constructs.

### 2.2. Measures

Methodology, study materials, raw and cleaned data, codes and translations are shared in The Open Science Framework (OSF) repository accessible to all teams contributing to the global database [36]. In the current study, we report results for the constructs described below (Appendix A).

#### 2.2.1. Outcome Variables

PHS behavior was assessed using the constructs Physical Contact, Physical Hygiene and Anti-Corona Policy Support, which were developed as ad hoc scales by leading authors of ICSMP (e.g., *Staying at home as much as practically possible, Keeping physical distance from all other people outside my home, Always washing my hands immediately after returning home, Disinfecting frequently used objects, such as mobile phones and keys, In favor of forbidding all public gatherings where many people are gathered at one place (sports and culture, “In favor of forbidding all non-necessary travel*). The measurement was on an 11-point scale (0—Strongly Disagree, 10—Agree).

#### 2.2.2. Predictor Variables

Conspiracy Theories of COVID-19 were assessed by the same point scale from 0—Strongly disagree to 10—Strongly agree (e.g., *The coronavirus is a bioweapon engineered by scientists, The coronavirus is a conspiracy to take away citizen’s rights for good and establish an authoritarian government, The coronavirus is a hoax invented by interest groups for financial gains*).

COVID-19 Risk Perception was assessed by answering the questions: “*By 30 April 2021: How likely do you think it is that you will get infected by the Coronavirus?*”, “*By 30 April 2021: How likely do you think it is that the average person in Bulgaria will get infected by the Coronavirus?*” (0% = Impossible, 50% = Neither likely nor unlikely, 100% = Certain).

Identity and Social Attitudes were represented by the scales National Identification, Collective Narcissism and Social Belonging. National Identification was assessed using two statements*:* “*I identify as Bulgarian*”, “*Being a Bulgarian is an important reflection of who I am*”. Collective Narcissism included the self-assessment of items: *“Bulgarians deserves special treatment*”, “*Not many people seem to fully understand the importance of Bulgarians*”, “*I will never be satisfied until Bulgarians gets the recognition it deserves*”. In terms of Social Belonging, the following items were used: “*I feel connected with others*”, “*When I am with other people*”, “*I feel included*”, “*I feel accepted by others*”, “*I have close bonds with family and friends*” (0—Strongly Disagree, 10—Agree).

Psychological well-being was examined by two items: “*To what extent you feel happy these days?*” (0—Very unhappy, 10—Very happy). Additionally, the second one was: “*Please imagine a ladder, with steps numbered 0 at the bottom and 10 at the top. The top represents the best possible life for you, and the bottom represents the worst possible life for you. On which step of the ladder would you say you personally feel you stand at this time?*” (0—Worst possible life, 10—Best possible life).

Moral Beliefs and Motivation were explored with the Morality-as-Cooperation and Moral Identity scales. The first scale included answers to the question “*When you decide whether something is right or wrong, to what extent are the following considerations relevant to your thinking?*” with items: “*Whether or not someone helped a member of their family*”, “*Whether or not someone worked to unite a community*”, “*Whether or not someone kept their promise*”, “*Whether or not someone showed courage in the face of adversity*”, “*Whether or not someone deferred to those in authority*”, “*Whether or not someone kept the best part for themselves*”, “*Whether or not someone kept something that didn’t belong to them*” (0—Strongly Disagree, 10—Agree). The assessment of Moral Identity was related to a visualization: “*…caring, compassionate, fair, friendly, generous, helpful, hardworking, honest, kind. The person with these characteristics could be you or it could be someone else. Visualize in your mind the kind of person who has these characteristics. Imagine how that person would think, feel, and act*”. Here, were added items, e.g., *It would make me feel good to be a person who has these characteristics, The types of things I do in my spare time* (e.g., *hobbies) clearly identify me as having these characteristics, Having these characteristics is not really important to me* (0—Strongly Disagree, 10—Agree).

The personality traits Open-mindedness, Trait Optimism, Trait Self-Control and Narcissism were examined. The items reflected tolerance to learning and mistakes (e.g., *I feel no shame learning from someone who knows more than me, Only wimps admit that they’ve made mistakes*), optimistic/pessimistic attitudes (e.g., *Overall, I expect more good things to happen to me than bad*), self-control skills (e.g., *I am good at resisting temptation, I have a hard time breaking bad habits*) and narcissistic characteristics (e.g., *I manage to be the center of attention with my outstanding contributions, Most people are somehow losers*) (0—Strongly Disagree, 10—Agree).

Some of the scales contained reversed items. In the Bulgarian adaptation of the study, we kept the assessment range from 0 to 10 points for all scales. According to us, this is closer to our ethnocultural attitude. After reviewing the raw data, we transformed the obtained scores into a five-point Likert scale so that neutral ratings would be interpreted as more refined.

#### 2.2.3. Confounding Variables

We collected quantitative and qualitative data on age, sex, marital status, number of children and employment status.

### 2.3. Statistical Analysis

We used two complementary approaches to establish the relative importance of different participant psychosocial characteristics as predictors of the Physical Contact, Physical Hygiene and Anti-Corona Policy Support scales. The first analytic step was to assess data distribution and distributional assumptions (Appendix A), reliability and bivariate relations between variables. We approached the data agnostically using a random forest machine learning algorithm to empirically identify which predictors contribute most to explaining COVID-19 Beliefs and Compliance variables. The algorithm uses 50% of the data in the machine learning sample of each tree (maximum number of decision trees, 100). For validation and testing of the algorithm, 20% of cases in the sample were used. Model evaluation was performed using mean squared error (MSE), root mean squared error (RMSE) and R2 indicators. The predictor importance was assessed by plotting the indicators’ mean decrease in accuracy and total increase in node purity, with higher values indicating greater impact of the predictor. As the next and main analysis step, we used linear regression models to test the significance of the predictors. All predictors were tested as independent variables simultaneously, the effect of each being adjusted for the influence of the others. Tests for multicollinearity between the predictors showed that there was no reason for concern (VIF < 5 and Tolerance index > 0.2) and they could be tested simultaneously. The analysis sample size was lower for these models (N = 615) due to missing data. A statistical significance level of *p* < 0.05 was adopted. All analyses were performed with SPSS Version 28.0 and JASP Version 0.17.1 [38,39].

## 3. Results

Cronbach’s Alpha coefficient was >0.700 for most scales, according to the generally accepted interpretation for reliability (Table 1). We refrain from interpreting Cronbach’s Alpha values < 0.700 as a sign of low reliability, since concrete scales are composed of a small number of items [40,41,42]. Here, we refer to the Physical Contact, National Identification, Morality-as-Cooperation and Open-mindedness scales. Breadth or narrowness of the construct measured can impact the scale’s reliability coefficient [43].

From Table 2, the sample is unbalanced and not representative of the general population. Most participants were female (67.3% females vs. 31.5% males) with an average age of 31.8 years (SD = 11.66). The majority were in a committed relationship and employed full-time.

Table 3 shows correlations between the variables in the study. Two polar trends regarding public health support were observed. The behavior of compliance to all measures of physical distance, hygiene and the recommended social-distancing policies was positively associated with higher Open-mindedness, Moral Identity and Risk Perception. Psychological well-being was strongly associated with contact avoidance and support for restrictive policies. Maintaining physical hygiene also positively correlated with Collective Narcissism, Morality-as-Cooperation, Trait Optimism, Social Belonging and Trait Self-Control. On the other hand, conspiracy beliefs were associated with lower compliance to public health measures, lower Risk Perception and with narcissistic traits and attitudes. Conspiracy beliefs were inversely related to Open-mindedness, but were associated with Morality-as-cooperation, optimism and self-control.

### Main Analyses

The data were approached agnostically using a random forest machine learning algorithm to empirically identify which predictors contribute most to explaining COVID-19 Beliefs and Compliance variables. Results of the random forest models exploring the contribution of participant characteristics to the outcomes are shown in Figure 1. Belief in conspiracy theories was the most influential predictor of all outcomes. Moral Identity also ranked relatively high as a predictor of Physical Hygiene and Anti-Corona Policy Support.

Next, we tested these observations with multivariate regressions, as shown in Table 4, Table 5 and Table 6. Physical Contact was explained at 18%, with several variables as significant predictors. Specifically, fewer Conspiracy Theories Beliefs, higher Collective Narcissism, Open-mindedness, higher Trait Self-Control, Moral Identity, Risk Perception and Psychological Well-Being were associated with higher levels of Physical Contact. Women reported higher Physical Contact scores than men (Table 4).

A similar trend was observed for Physical Hygiene, where Conspiracy Theories Beliefs related to lower Physical Hygiene, while higher Collective Narcissism, Moral Identity and Psychological Well-Being were related to better Physical Hygiene (Table 5). Women and participants in a relationship reported higher Physical Hygiene than men and those who were single. This model explained 14% of the variance in Physical Hygiene.

In the third model, 25% of the variance in Anti-Corona Policy Support was explained. Once again, Conspiracy Theories were significantly associated with weaker Anti-Corona Policy Support, while higher Collective Narcissism, Moral Identity, Risk Perception, Psychological Well-Being and being a woman were associated with stronger Anti-Corona Policy Support.

## 4. Discussion

Two polar trends in the public health behavior of Bulgarians were observed. The first trend of public health non-support revealed more complex dependencies. Conspiracy beliefs were a significant predictor of lower compliance with public health anti-epidemic measures, as was found in most countries included in the global survey [26]. On the one hand, Conspiracy Theories Beliefs were significantly associated with lower Risk Perception and lower levels of Open-mindedness, and on the other hand were associated with the Morality-as-Cooperation, Trait Optimism and Trait Self-Control dimensions. Such an affective ratio in behavior has been interpreted as a moral dilemma state by researchers [27,42]. This condition resembles a value conflict in following policies that people distrust but are able to co-relate ethically. These constructs are related to cooperativeness, Self-directedness and Self-transcendence as adaptive personality traits in the psychobiological model [34,39].

Multivariate regressions confirmed the two distinct patterns of support and non-support. All PHS dimensions were consistently and negatively predicted by Conspiracy Theories Beliefs. Collective Narcissism, Moral Identity and Psychological Well-Being were consistent predictors of PHS dimensions. These findings are a marker of affective polarization. They can also be seen as socio-cognitive polarization, as was described in a recent study [10]. The data were generated at the time of the first lockdown, when people were confined to their homes. It was a collective phase of anxiety and resistance. Inhibited psychosocial functioning was transformed into affective and adaptive behaviors. Relations were overloaded by virtual communications, and analysts of psychological effects of the pandemic used concepts such as cognitive invasion, cognitive acceleration and sensory deprivation [44]. In all prediction models, women were associated with significantly higher support scores.

The second trend of public health support was significantly associated with Open-mindedness, Moral Identity and Risk Perception. The compliance with physical hygiene correlated positively with Collective Narcissism, Morality-as-Cooperation, Trait Optimism, Trait Self-Control and Social Belonging. Establishment of these dependencies may be related to rational behavior of acceptance and awareness to the realities of the pandemic. Moreover, they are indicative of value-oriented behavior [28,29].

A particularly important finding was the significant association between Psychological Well-Being, avoidance of Physical Contact and Anti-Corona Policy Support. Well-being is an implicit characteristic of human functioning. Subjective well-being reflects cognitive and emotional aspects of the experience of individual life circumstances [25]. In other words, the affective reprocessing of human experience is a phenomenon of psychological well-being. Personality profiles in terms of Robert Cloninger’s model are considered to be among the most consistent predictors of well-being because they specify the synergistic nonlinear relations between emotion and cognition. For instance, the combination of high Self-directedness, Cooperativeness and Self-transcendence (the three TCI character dimensions) predicts greater physical, mental and social well-being than any other profile or trait [45].

Previous studies have researched a more in-depth interpretation of subjective well-being using a model of affective profiles [46,47]. Four affective profiles have been defined: individuals who are self-fulfilling (high positive affect, low negative affect), individuals who are highly affective (high positive affect, high negative affect), individuals who are low affective (low positive affect, low negative affect) and individuals who are self-destructive (low positive affect, high negative affect). Various results have been reported based on that profiling, e.g., that individuals with self-fulfilling and highly affective profiles perform best during stressful situations and demonstrate a more dynamic lifestyle than low affective and self-destructive individuals. Self-fulfilling individuals also believe that they are more energetic and optimistic and indicate greater life satisfaction and psychological well-being compared with individuals with the other affective profiles. Individuals with self-fulfilling profiles are characterized by high self-esteem, high optimism and an internal locus of control, whereas individuals with self-destructive profiles have inherently low self-esteem, low optimism and an external locus of control. There is evidence that self-destructive and highly affective profiles are more strongly associated with more severe post-traumatic stress disorder symptoms in Dutch victims of violence, as well as evidence from cross-cultural comparative studies that report differences in life satisfaction and psychological well-being [46,47,48,49]. Although the research methodology does not capture the structure of well-being in this study, the correlations between psychological well-being, contact avoidance and agreement with anti-corona policies can be interpreted as evidence of an affective mode of adaptive functioning.

The time of conducting the research preceded the first epidemiological wave of illness from coronavirus in Bulgaria [50]. It was the period of the first long-term total lockdown in Bulgaria, when socio-economic life was reduced to distance education and work. People were facing a looming economic collapse and potential job loss, and stress and anxiety levels were very high [51,52,53]. The adaptation to an unpredictable duration of the pandemic was in a process of active psychologization. We argue that trends of support and non-support can be seen phenomenologically as processes of precarity and non-precarity in pandemic realities. Conspiracy mentality is motivated by a basic sense of distrust and ontological uncertainty [3,4,5], and the recent pandemic created events and psychosocial contexts that activated existential emotions in people. Our results align with an affective framework of adaptation expressed in the support and non-support of public health, and those should be observed seriously in action plan protocols as a major resource to foster cooperativeness and resilience by avoiding aggressive and self-contradictory measures and by means of increased awareness and respect of the health attitudes of a specific population.

Our results can inform and motivate more careful, consistent with attitudes and evidence-based decision-making under the conditions of a similar public health crisis to limit the collateral damages of the pandemic both in terms of economic burden and increased anxiety and worries on the population level [44,53,54,55].

## 5. Limitations

This study was not representative of the general population in terms of sex, age, ethnicity and educational structure. It was conducted before the inclusion of (obligatory) vaccination into the public health policies to prevent the spread of COVID-19 at a population level. This particular intervention in the territory of shared decision making on one hand and the privacy of individual informed consent present a major ethical concern as a determinant of public health policy support after the first wave of the pandemic and is not considered in our design.

A narrower range of constructs due to the availability of the global data set were focused on. They are part of a constellation of relevant predictors. Furthermore, an agnostic approach to explain maximum variation in the outcome variables was adopted. Therefore, it is recommend that future studies test hypotheses regarding the interrelationships of these constructs through structural modeling, which were not considered due to the exploratory nature of the analyses and the fact that, with cross-sectional data, it is preferable to be cautious about assuming causality between the predictors themselves.

## 6. Conclusions

This study revealed two polar trends in the public health behavior response of Bulgarians during the outbreak of the COVID-19 crisis. The tendency not to support public health was significantly predicted by the presence of beliefs in conspiracy theories. The trend of PHS was significantly associated with Open-mindedness, Moral Identity and Risk Perception. Those results outline a values-based profile of the initial response to the critical situation in Bulgaria, which, however, was later distorted by inconsistent health policies and decision making during the following waves of the pandemic and by mandatory vaccination. The social and affective polarities composed of conspiracy beliefs undermined public health support during the crisis and contributed to the wide spread of antagonistic speculations in society. In effect, such antagonism probably leads to less cooperation with anti-epidemic measures and vaccination policies and high mortality rates at the population level. The take-home messages from our study may be incorporated into guidelines to provide more coherent public health policy, which may secure the adaptive behavior and compliance at the population level and thereby limit the direct and indirect burden from similar crises in the future.

## Figures and Tables

**Figure 1 brainsci-13-00821-f001:**
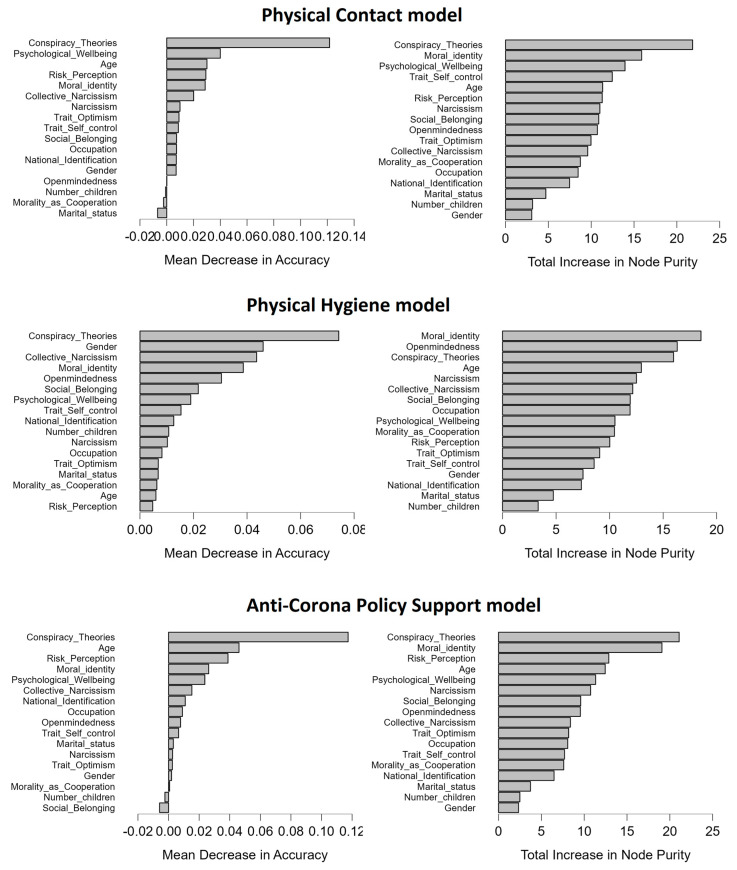
Variable importance plots from random forest models predicting Physical Contact (top row; R^2^ = 13%), Physical Hygiene (middle row; R^2^ = 14%) and Anti-Corona Policy Support (bottom row; R^2^ = 20%). The ranking of variable importance is based on mean decrease in accuracy and total increase in node purity, with higher values indicating greater impact of the predictor.

**Table 1 brainsci-13-00821-t001:** Psychometric statistics of the psychological constructs used in the study.

Scale	Psychosocial Dimensions	Mean	±SD	Cronbach’s α	No. of Items	Item–ITEM Correlations (Min–Max)	Corrected Item–Total Correlations (Min–Max)
PC	Physical Contact	20.12	±3.81	0.684	5	0.152–0.622	0.268–0.595
PH	Physical Hygiene	21.02	±4.05	0.756	5	0.257–0.874	0.422–0.684
ACPS	Anti-Corona Policy Support	19.56	±5.08	0.859	5	0.438–0.767	0.536–0.772
CN	Collective Narcissism	8.15	±3.45	0.838	3	0.612–0.632	0.684–0.716
PWB	Psychological Well-Being	6.19	±1.74	0.769	2	0.629	0.629
CT	Conspiracy Theories COVID-19	9.73	±4.83	0.893	4	0.556–0.783	
NI	National Identification	8.23	±1.89	0.550	2	0.429	0.429
MC	Morality-as-Cooperation	24.65	±3.07	0.339	7	−0.370–0.566	−0.320–0.444
OM	Open-mindedness	26.03	±3.02	0.561	6	0.066–0.516	0.155–0.479
TO	Trait Optimism	7.91	±1.70	0.833	2	0.714	0.714
SB	Social Belonging	15.92	±2.74	0.778	4	0.368–0.590	0.489–0.653
TSC	Trait Self-Control	14.03	±2.79	0.577	4	0.174–0.375	0.311–0.421
N	Narcissism	15.48	±4.73	0.759	6	0.115–0.586	0.264–0.637
MI	Moral Identity	40.23	±7.72	0.772	10	−0.104–0.691	0.235–0.643
RP	Risk Perception	6.07	±1.86	0.752	2	0.604	0.604

**Table 2 brainsci-13-00821-t002:** Participant characteristics in the study.

Demographics	Respondents (N = 733)
	Mean ± SD
**Age**	31.81 ± 11.66
	N (%)
**Sex**	
Female	493 (67.3%)
Male	231 (31.5%)
Other	4 (0.5%)
No answer specified	5 (0.7%)
**Marital status**	
Single	276 (37.7%)
In a relationship	269 (36.7%)
Married	175 (23.9%)
No answer specified	13 (1.7%)
**Children**	
One	128 (17.5%)
Two	115 (15.7%)
Three	9 (1.2%)
Four	1 (0.1%)
Five	2 (0.3%)
None	465 (63.4%)
No answer specified	13 (1.8%)
**Occupation**	
Employed full time	302 (41.2%)
Employed part-time	45 (6.1%)
Unemployed/Looking for work	42 (5.7%)
Student	164 (22.4%)
Retired	4 (0.5%)
Other	153 (20.9%)
No answer specified	23 (3.2%)

**Table 3 brainsci-13-00821-t003:** Correlations between COVID-19 Beliefs and Compliance, Identity and Social Attitudes, Psychological well-being, Moral Beliefs, Motivation and Personality Traits dimensions.

	PC	PH	ACPS	CN	PWB	CT	NI	MC	OM	TO	SB	TSC	N	MI	RP
PC	-	**0.407 ****	**0.585 ****	−0.031	**0.108 ****	**−0.282 ****	−0.014	0.012	**0.196 ****	0.028	0.056	**0.101 ****	**−0.104 ****	**0.153 ****	**0.161 ****
PH		-	**0.477 ****	**0.112 ****	−0.058	**−0.093 ***	0.073	**0.131 ****	**0.100 ****	**0.112 ****	**0.155 ****	**0.124 ****	−0.001	**0.224 ****	**0.117 ****
ACPS			-	0.020	**0.171 ****	**−0.330 ****	−0.018	0.039	**0.155 ****	0.032	**0.073 ***	0.037	−0.048	**0.179 ****	**0.201 ****
CN				-	−0.052	**0.370 ****	**0.412 ****	**0.170 ****	**−0.165 ****	**0.166 ****	**0.174 ****	**0.135 ****	**0.201 ****	**0.156 ****	0.000
PWB					-	**−0.143 ****	0.011	**−0.075 ***	**0.086 ***	**0.159 ****	−0.010	−0.015	0.005	−0.011	0.040
CT						-	**0.211 ****	**0.081 ***	**−0.180 ****	**0.129 ****	0.058	**0.107 ****	**0.135 ****	**0.086 ***	**−0.092 ***
NI							-	**0.187 ****	0.033	**0.244 ****	**0.274 ****	**0.168 ****	0.070	**0.230 ****	0.014
MC								-	0.026	**0.139 ****	**0.185 ****	**0.129 ****	0.047	**0.222 ****	0.022
OM									-	−0.009	**0.132 ****	0.029	**−0.279 ****	**0.090 ***	**0.114 ****
TO										-	**0.404 ****	**0.364 ****	0.024	**0.191 ****	−0.045
SB											-	**0.214 ****	0.028	**0.371 ****	0.068
TSC												-	**−0.117****	**0.268****	0.041
N													-	**0.167****	−0.023
MI														-	0.039
RP															-

** *p* < 0.01, * *p* < 0.05.

**Table 4 brainsci-13-00821-t004:** Predictors of Physical Contact (N = 615).

Predictors	β	*p*	95% CI	
			Lower	Upper
(Intercept)	12.468	<0.001	8.190	16.745
**Conspiracy Theories**	**−0.233**	**<0.001**	**−0.297**	**−0.169**
Age	−0.027	0.179	−0.065	0.012
Number of children	−0.098	0.717	−0.629	0.433
**Collective Narcissism**	**0.111**	**0.025**	**0.014**	**0.208**
National Identification	−0.107	0.236	−0.283	0.070
**Open-mindedness**	**0.135**	**0.009**	**0.034**	**0.236**
Morality-as-Cooperation	0.011	0.814	−0.084	0.107
Trait Optimism	0.023	0.820	−0.173	0.218
Social Belonging	−0.046	0.458	−0.166	0.075
**Trait Self-Control**	**0.125**	**0.030**	**0.012**	**0.237**
Narcissism	−0.059	0.069	−0.123	0.005
**Moral Identity**	**0.078**	**<0.001**	**0.037**	**0.119**
**Risk Perception**	**0.225**	**0.004**	**0.072**	**0.378**
**Psychological Well-Being**	**0.251**	**0.004**	**0.081**	**0.422**
**Gender (woman vs. man)**	**0.818**	**0.012**	**0.183**	**1.452**
Marital status (single)	Ref.			
Marital status (in a relationship)	−0.458	0.171	−1.114	0.198
Marital status (married)	0.300	0.545	−0.673	1.274
Occupation (full-time employee)	Ref.			
Occupation (part-time employee)	−0.344	0.584	−1.574	0.887
Occupation (unemployed)	0.709	0.255	−0.513	1.931
Occupation (student)	−0.142	0.746	−0.999	0.715
Occupation (retired)	−0.571	0.825	−5.633	4.491
Occupation (other)	0.804	0.046	0.013	1.594

Coefficients shown are unstandardized linear regression coefficients with corresponding significance level and 95% confidence interval (95% CI). Model Adjusted R^2^ = 0.1.

**Table 5 brainsci-13-00821-t005:** Predictors of Physical Hygiene (N = 615).

Predictors	β	*p*	95% CI	
			Lower	Upper
(Intercept)	11.546	< 0.001	6.965	16.126
**Conspiracy Theories**	**−0.170**	**<0.001**	**−0.238**	**−0.101**
Age	0.005	0.799	−0.036	0.047
Number of children	−0.527	0.069	−1.094	0.041
**Collective Narcissism**	**0.134**	**0.011**	**0.031**	**0.238**
National Identification	−0.090	0.347	−0.279	0.098
Open-mindedness	0.072	0.188	−0.035	0.180
**Morality-as-Cooperation**	**0.104**	**0.046**	**0.002**	**0.206**
Trait Optimism	0.178	0.093	−0.030	0.387
Social Belonging	0.060	0.362	−0.069	0.189
Trait Self-Control	−0.018	0.768	−0.138	0.102
Narcissism	−0.012	0.733	−0.080	0.057
**Moral Identity**	**0.092**	**<0.001**	**0.048**	**0.136**
Risk Perception	0.123	0.139	−0.040	0.286
**Psychological Well-Being**	**−0.264**	**0.005**	**−0.445**	**−0.082**
**Gender (woman vs. man)**	**1.383**	**<0.001**	**0.704**	**2.062**
Marital status (single)	Ref.			
**Marital status (in a relationship)**	**0.802**	**0.025**	**0.100**	**1.504**
**Marital status (married)**	**1.179**	**0.026**	**0.139**	**2.219**
Occupation (full-time employee)	Ref.			
Occupation (part-time employee)	0.282	0.674	−1.035	1.598
Occupation (unemployed)	0.156	0.814	−1.151	1.464
Occupation (student)	0.381	0.417	−0.540	1.301
Occupation (retired)	−3.445	0.212	−8.860	1.971
Occupation (other)	0.361	0.406	−0.491	1.213

Coefficients shown are unstandardized linear regression coefficients with corresponding significance level and 95% confidence interval. Model Adjusted R^2^ = 0.14.

**Table 6 brainsci-13-00821-t006:** Predictors of Anti-Corona Policy Support (N = 615).

Predictors	β	*p*	95% CI	
			Lower	Upper
(Intercept)	10.095	<0.001	4.584	15.607
**Conspiracy Theories**	**−0.406**	**<0.001**	**−0.489**	**−0.324**
Age	−0.037	0.143	−0.087	0.013
Number of children	0.402	0.248	−0.280	1.085
**Collective Narcissism**	**0.221**	**<0.001**	**0.096**	**0.346**
National Identification	−0.140	0.228	−0.368	0.088
Open-mindedness	0.099	0.136	−0.031	0.229
Morality-as-Cooperation	0.013	0.833	−0.110	0.137
Trait Optimism	0.065	0.611	−0.187	0.317
Social Belonging	0.001	0.985	−0.153	0.156
Trait Self-Control	−0.036	0.629	−0.180	0.109
Narcissism	−0.039	0.351	−0.122	0.043
**Moral Identity**	**0.130**	**<0.001**	**0.077**	**0.183**
**Risk Perception**	**0.382**	**<0.001**	**0.185**	**0.580**
**Psychological Well-being**	**0.480**	**<0.001**	**0.261**	**0.699**
**Gender (woman vs. man)**	**0.998**	**0.017**	**0.182**	**1.815**
Marital status (single)	Ref.			
Marital status (in a relationship)	−0.294	0.495	−1.140	0.552
Marital status (married)	−0.278	0.663	−1.530	0.974
Occupation (full-time employee)	Ref.			
Occupation (part-time employee)	−0.147	0.856	−1.733	1.439
Occupation (unemployed)	0.807	0.315	−0.767	2.381
Occupation (student)	0.836	0.138	−0.268	1.940
Occupation (retired)	0.107	0.974	−6.419	6.632
Occupation (other)	0.434	0.404	−0.586	1.455

Coefficients shown are unstandardized linear regression coefficients with corresponding significance level and 95% confidence interval. Model Adjusted R^2^ = 0.24.

## Data Availability

The data presented in this study are available on request from the corresponding author.

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
