# Peer review of "Retrospective Analysis of the Psychological Predictors of Public Health Support in Bulgarians at the Beginning of the Coronavirus Pandemic"

_brainsci, 2023, doi:10.3390/brainsci13050821_

Round 1

Reviewer 1 Report

The study has very interesting aspects regarding Covid-19 and I think the revision should be minimal.

Starting with the keywords, I would not use the word "Prediction" but "Psychological Predictors" as in the title. Be careful, some words are written in capital letters, others in lower case, with no consistency (e.g., Conspiracy Theories Beliefs, I would put Beliefs in capital letters. This is also in the text).

As far as the theoretical background is concerned, I think that more research is needed, including other studies on Covid-19 and similar topics.

I recommend these papers, for example:

-          Gennaro, A., Reho, M., Marinaci, T., Cordella, B., Castiglioni, M., Caldiroli, C. L., & Venuleo, C. (2023). Social Environment and Attitudes toward COVID-19 Anti-Contagious Measures: An Explorative Study from Italy. International Journal of Environmental Research and Public Health20(4), 3621.

-          Galende, N., Redondo, I., Dosil-Santamaria, M., & Ozamiz-Etxebarria, N. (2022). Factors Influencing Compliance with COVID-19 Health Measures: A Spanish Study to Improve Adherence Campaigns. International Journal of Environmental Research and Public Health19(8), 4853.

-          Lorettu, L., Mastrangelo, G., Stepien, J., Grabowski, J., Meloni, R., Piu, D., ... & Cegolon, L. (2021). Attitudes and perceptions of health protection measures against the spread of COVID-19 in Italy and Poland. Frontiers in psychology12, 805790.

-          Negri, A., Conte, F., Caldiroli, C. L., Neimeyer, R. A., & Castiglioni, M. (2023). Psychological Factors Explaining the COVID-19 Pandemic Impact on Mental Health: The Role of Meaning, Beliefs, and Perceptions of Vulnerability and Mortality. Behavioral Sciences13(2), 162.

-          Nwakasi, C., Esiaka, D., Uchendu, I., & Bosun-Arije, S. (2022). Factors influencing compliance with public health directives and support for government's actions against COVID-19: A Nigerian case study. Scientific African15, e01089.

-          Xie, Q., Sundararaj, V., & Mr, R. (2022). Analyzing the factors affecting the attitude of public toward lockdown, institutional trust, and civic engagement activities. Journal of community psychology50(2), 806-822. 

Beware of acronyms. For example, PHS is used first in the abstract and later in the text on line 114 and, after, on line 153.

On line 114 it is not written in full (Public health support), but on line 153 it is. You would consider writing it in full the first time and not later. Or at least write it in full on line 114, as this is the first time it appears in the text.

In the "Materials and methods" section, you would try to make it clearer what goes in paragraph 2.2 and what has to go in paragraph 2.3.

The conclusions need to be rewritten in a way that is appropriate for a scientific article; I think the article provides a lot of material to think about and write with. Perhaps, the initial theoretical background on covid-19 will also help.

Moderate editing of English language

Reviewer 2 Report

First, I would like to thank the authors of the manuscript “Retrospective analysis of the Psychological Predictors of Public 2 Health Support in Bulgarians at the beginning of the corona- 3 virus pandemic” for presenting the results of their study.

The study examined the earliest critical context of the COVID-19 pandemic in Bulgaria before the first epidemiological wave of the contagion. It adopted a retrospective and agnostic analytical approach and aimed to identify traits and trends that explain the public health support (PHS) of Bulgarians during the first two months of the declared state of emergency. The study used a set of variables with a unified method within an international scientific network named International Collaboration on Social & Moral Psychology of COVID-19 (ICSMP) in April and May 2020.

The results of the study revealed that conspiracy theories beliefs were a significant predictor of lower PHS. Psychological well-being was significantly associated with physical contact and anti-corona policy support. Physical contact was significantly predicted by less conspiracy theories beliefs, higher collective narcissism, open-mindedness, higher trait self-control, moral identity, risk perception, and psychological well-being. Physical hygiene compliance was predicted by less conspiracy theories beliefs, collective narcissism, morality-as-cooperation, moral identity, and psychological well-being. The study found two polar trends of support and non-support of public health policies, which suggest affective polarization and the phenomenology of (non)precarity during the outbreak of the pandemic.

Overall, this study contributes to the understanding of the socio-affective perspective of the COVID-19 pandemic in Bulgaria and provides evidence for affective polarization and the phenomenology of (non)precarity during the outbreak of the pandemic.

Title: Adequate.

Abstract: Specify in brief the sample size % of female and age mean and SD.

Introduction: Adequate. Relevant.

Methods: The measures section describes the variables considered and how they were analysed, but does not give details on the survey instruments origins. The authors could provide more detailed information about the grid or standardised questionnaires (survey, test, scale?) or especially designed by Authors used to collect the data in the measures section.

Data analysis:  Adequate.

Discussion: Adequate.

Figures & Tables: Adequate

References: Adequate

I hope I have made some insightful suggestions that can enhance your study,

good luck!

Reviewer 3 Report

Thank you for the opportunity to revise this manuscript. The paper aims at

Identifying, through a retrospective analysis, traits and trends that explain public health support (PHS) of Bulgarians during the first two months of the declared COVID-19 emergency.

To this aim, the authors used a very large number of predictors, that they presented in the Introduction. In my opinion, one of most significant limitation of the paper is the Introduction, that sounds a little bit confusing. It is not clear on which basis the authors chose to select some variables. Sure, many other variables could have been chosen as possible predictors of PHS. It is not clear the rational that lead to select these specific variables and the way they are interconnected one-another. I suggest the authors to be more clear in this regard and, if this is the case, to remove some variable.

Consistent with these changes, the final section of the Introduction should better clarify the authors’ expected results.

Line 76-79: please provide a reference for this sentence

Line 119-120: what do the large multinational sample refers to?

Line 134-135: what responses were excluded? Which were the inclusion/exclusion criteria?

Measures section: the authors should provide example items for every predictor variable. Moreover, they should provide a reference for the tools

Line 221: a reference is needed

Figure 1: The authors could add also in the figure the names of models (i.e., Physical Contact, Physical Hygiene and Anti-Corona Policy Support).

A section regarding the practical implications of the study should be expanded.

Overall English language needs a proofreading.

Round 2

Reviewer 3 Report

The revised version of the manuscript is suitable for publication

Congratulations to the authors

Author Response

Dear Academic Editor,

thank you for your constructive suggestion, which has been implemented in due course.

Please see the revised version uploaded herewith.

Best regards,

Kristina Stoyanova
